# M2T2: Multi-Task Masked Transformer for Object-centric Pick and Place

**Wentao Yuan**
University of Washington
wentaoy@cs.washington.edu

**Adithyavairavan Murali** *
NVIDIA
admurali@nvidia.com

**Arsalan Mousavian** *
NVIDIA
amousavian@nvidia.com

**Dieter Fox**
University of Washington, NVIDIA
fox@cs.washington.edu

**Abstract:** With the advent of large language models and large-scale robotic datasets, there has been tremendous progress in high-level decision-making for object manipulation [1, 2, 3, 4]. These generic models are able to interpret complex tasks using language commands, but they often have difficulties generalizing to out-of-distribution objects due to the inability of low-level action primitives. In contrast, existing task-specific models [5, 6] excel in low-level manipulation of unknown objects, but only work for a single type of action. To bridge this gap, we present M2T2, a single model that supplies different types of low-level actions that work robustly on arbitrary objects in cluttered scenes. M2T2 is a transformer model which reasons about contact points and predicts valid gripper poses for different action modes given a raw point cloud of the scene. Trained on a large-scale synthetic dataset with 128K scenes, M2T2 achieves zero-shot *sim2real* transfer on the real robot, outperforming the baseline system with state-of-the-art task-specific models by about 19% in overall performance and 37.5% in challenging scenes were the object needs to be re-oriented for collision-free placement. M2T2 also achieves state-of-the-art results on a subset of language conditioned tasks in RLBench [7]. Videos of robot experiments on unseen objects in both real world and simulation are available on our project website.

**Keywords:** Object Manipulation, Pick-and-place, Multi-task Learning

## 1 Introduction

The successful completion of many complex manipulation tasks such as object rearrangement relies on robust action primitives that can handle a large variety of objects. Recently, tremendous progress has been made in open-world object manipulation [1, 2, 3, 4] using language models for high-level planning. However, these methods are often restricted to scenes with a few fixed object shapes due to the limited capability of low-level skills such as picking and placing. Meanwhile, there are task-specific models [5, 8, 9] that excel on a particular skill on a large variety of objects. This leads us to the question: is it possible to have a single model for different action primitives that works robustly on diverse objects?

We propose **M**ulti-**T**ask **M**asked **T**ransformer (M2T2), a unified model for learning multiple action primitives. As shown in Fig. 1, given a point cloud of the scene, M2T2 predicts collision-free gripper poses for various types of actions including 6-DoF grasping and placing, eliminating the need to use different methods for different actions. M2T2 can generate a diverse set of goal poses that provide sufficient options for low-level motion planners. It can also generate more specific goal poses conditioned on language. Combining high-level task planners and the robust action primitives from M2T2 allows the robot to solve many complex tasks like the ones in RLBench [7]. Overall, our contributions are as follows:

7th Conference on Robot Learning (CoRL 2023), Atlanta, USA.

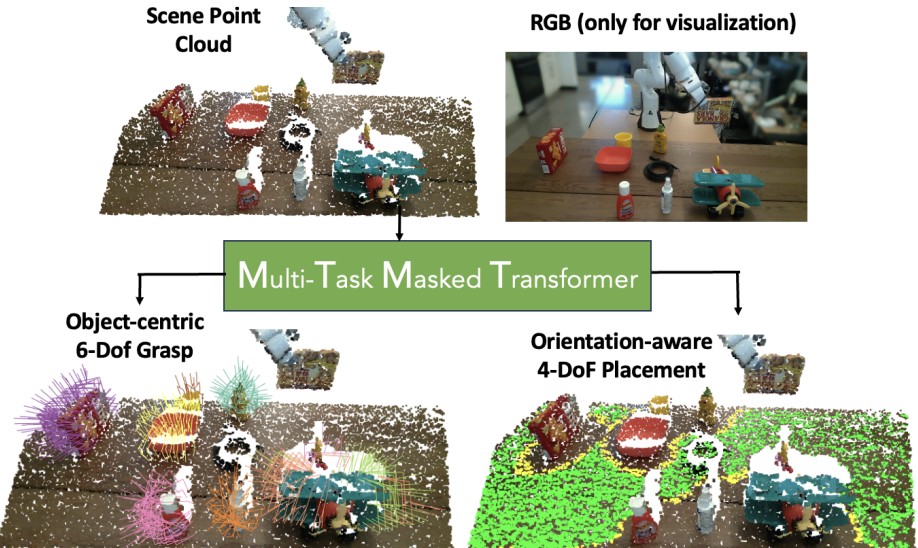

Figure 1: We propose M2T2, a unified model for learning multiple action primitives. M2T2 takes a raw 3D point cloud and predicts 6-DoF grasps per-object (lower left) and orientation-aware placements (lower right, where green means the object can fit in any orientation and yellow means only a subset of orientations are possible). Colors on the point clouds are for visualization only.

1. We present M2T2, a unified transformer model for grasping and placing, which outperforms state-of-the-art methods [5, 6] in terms of success rate and output diversity.

2. A large-scale synthetic dataset for training M2T2, consisting of 130K cluttered scenes with 8.8K different objects, annotated with valid gripper poses for picking and placing.

3. We show that M2T2 achieves zero-shot *sim2real* transfer for picking and placing out-of-distribution objects, outperforming baseline by about 19%.

4. We show that M2T2 outperforms state-of-the-art end-to-end method [10] on a subset of RLBench [7], demonstrating its potential in solving complex tasks with language goals.

## 2 Related Work

**Multi-Task Learning in Robotics**   With the advent of robotic datasets with diverse tasks [7, 11], many recent works have shown that learning multiple manipulation tasks with a single model can improve sample-efficiency and performance. Some works learn a common representation for multiple tasks [12, 13], while other works [2, 10, 14] train end-to-end language-conditioned policies via imitation learning. However, these end-to-end agents have a hard time generalizing to out-of-distribution tasks and objects. In contrast, we take a more modular approach. M2T2 supplies action primitives like "placing the mug" that work on unseen objects in the real world. By interfacing with other task planning modules [3, 4], M2T2 can be part of a robust and flexible open-world manipulation system.

**Object Grasping**   Grasping is the most fundamental skill of a robot manipulator. Recently, 6-DoF, learning-based grasp pose generator is becoming mainstream. These methods typically take a 3D point cloud [5, 8, 15, 16] or voxelization of the scene [17, 18] and predict 6-DoF gripper poses to stably grasp an object. There are also task-oriented grasp generators [19, 20] that predicts grasps for downstream tasks (e.g. handover). However, these works are geared toward a single skill, grasping. While grasping is an important skill, it cannot solve all the tasks in manipulation. M2T2 aim to build a common formulation for different kinds of skills, including grasping.

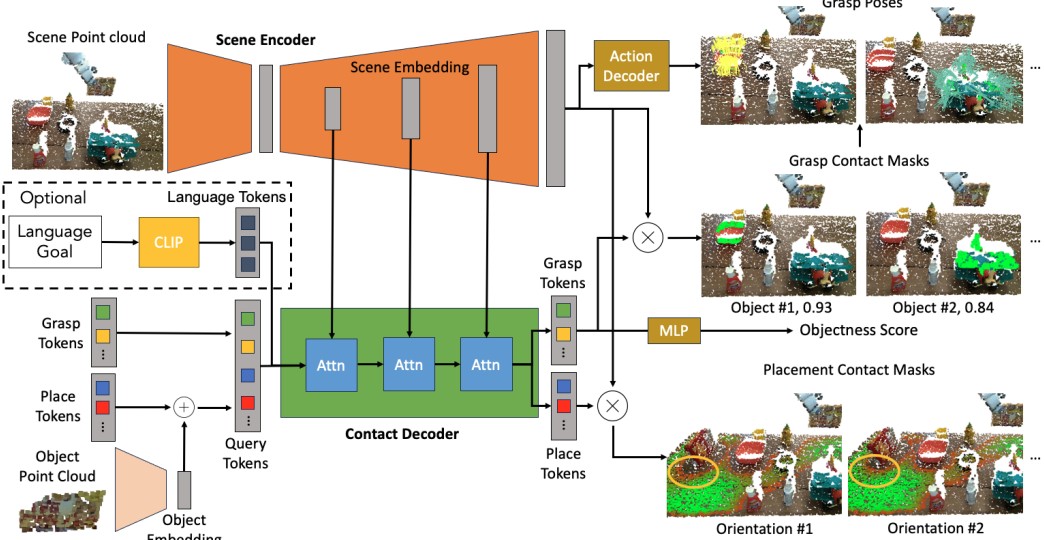

Figure 2: M2T2 generates valid gripper poses for grasping and placing with a single model. First, a 3D network (scene encoder) takes the scene point cloud and produces multi-scale feature maps. Then, the features are cross-attended with learnable query tokens via a transformer (contact decoder). Finally, the output tokens are multiplied with per-point features and generate contact masks and gripper poses for each object (for grasping) and each orientation (for placing). For grasping, addition MLPs are applied to the output tokens and per-point features to predict objectness scores (to filter out non-object proposals) and grasp parameters (to reconstruct gripper poses). Optionally, the contact decoder can take a set of tokens encoding language goals to produce goal-conditioned grasping and placing poses.

**Object Placement** Placement is another important action mode for a robot manipulator. Compared to grasping methods, placement is less studied until recently. [6] uses rejection sampling and learning-based collision detector [21] to find all possible placement positions. [22] generates placement configurations for a set of objects based on language commands, but it does not consider the gripper. M2T2 predicts placement poses without the need for sampling, uses the same model for grasping and considers both the position and orientation of the object and the gripper.

## 3 Technical Approach

M2T2 predicts target gripper poses for multiple action primitives. Here, we consider two most fundamental action modes of a robot manipulator, picking and placing.

**Object-centric 6-DoF Grasping:** The input is a single 3D point cloud of the scene which can be extracted from commodity depth sensors. The output is set of object grasp proposals. Each object grasp proposal is a set of 6-DoF grasp poses (3-DoF rotation + 3-DoF translation), indicating where the end effector of a robot arm needs to be in order to pick up the object successfully.

**Orientation-aware Placing:** The input is a 3D point cloud of the scene plus a partial 3D point cloud of the object to be placed. The output is a set of 6-DoF placement poses, indicating where the end effector needs to be so that when the object is released, it will be placed stably without collision. Note that M2T2 generates not only the location, but also the orientation of the object to be placed. This ensures that the object can be re-oriented before placed to fit into cluttered spaces.

The key idea of M2T2 is to reason about contact points. We view picking as the robot making contact with a target object using an empty gripper and placing as the robot using an object in its gripper to make contact with a surface.

## 3.1 Architecture

**Scene Encoder:** The scene encoder encodes a 3D point cloud of the scene into multi-scale feature maps that serve as context for the contact decoder. Specifically, it produces four feature maps that are $1/64, 1/16, 1/4, 1$ times the input size respectively. Each feature vector in the feature maps is grounded to a point in the input point cloud. We adapt a PointNet++ [23] designed for semantic segmentation as the scene encoder, but in principle, any network that produces multi-resolution feature maps from 3D point clouds can serve as the scene encoder.

**Contact Decoder:** The contact decoder is a transformer that predicts where to make contact for both grasping and placing. We used the grasp representation of [5] for grasping where each grasp is anchored around the visible point on the object that makes contact with the gripper during grasping and the model predicts additional parameters specifying the relative transform of grasp with respect to the contact point. We extend this representation for placing by defining contact point as the location where the center of object point cloud projects on the table.

As a result, we can borrow the latest insight from image segmentation. In our case, we modify the masked transformer [24] to predict contact masks. The transformer takes a set of learnable query tokens through multiple attention layers. Feature maps of multiple resolutions from scene encoder are passed in via cross-attention at different layers. The output tokens of each layer are multiplied with the per-point feature map from scene encoder to generate interim masks. The interim masks are used to mask the cross attention in the next layer to guide the attention into relevant regions (thus the name "masked transformer"). After the last attention layer, the model produces $G$ grasping masks and $P$ placing masks, where $G$ is the maximum number of graspable objects and $P$ is the number of placement orientations.

**Objectness MLP:** An MLP takes the grasp tokens and produces an objectness score for each token. This is to filter out the non-object tokens since the number of graspable objects in the scene can vary (see Sec. 3.2 and Sec. 3.3 for how the score is used in training and inference).

**Object Encoder:** The object encoder is a PointNet++ [23] which encodes a 3D point cloud of the object to be placed to a single feature vector that is added to the place query tokens.

**Action Decoder:** The action decoder is a 3-layer MLP that takes the per-point feature map from scene encoder and predicts a 3D approach direction, a 3D contact direction and a 1D grasp width for each point, which are used to reconstruct grasp poses together with the contact points (see Sec. 3.3).

## 3.2 Training Objective

**Grasping:** The grasping objective consists of three terms: objectness loss $L_{\text{obj}}$, mask loss $L_{\text{mask}}$ and ADD-S loss $L_{\text{ADD-S}}$.

Because the number of objects $N$ in the scene is unknown, we set $G$, the number of grasp tokens, to a large number (see Sec. 4.2 for ablations). M2T2 outputs $G$ scalar objectness scores $o_i$ and $G$ per-point masks $M_i^{\text{grasp}}$. We use Hungarian matching to select $N$ masks that best match with the ground truth. First, we compute the following cost for each prediction $(o_i, M_i^{\text{grasp}})$ and ground truth mask $M_j^{\text{gt}}$

$$C_{ij} = 1 - o_i + \text{BCE}(M_i^{\text{pred}}, M_j^{\text{gt}}) + \text{DICE}(M_i^{\text{pred}}, M_j^{\text{gt}}) \tag{1}$$

where BCE is binary cross entropy and DICE is the DICE loss [25]. Now, we apply Hungarian matching to the $G \times N$ cost matrix $C$ to obtain the set of indices $\mathcal{M} = \{m_i\}$ that minimizes the total cost $\sum_{j=1}^{N} C_{m_j j}$. Then, we compute the objectness loss by labeling all matched tokens as positive and others as negative

$$L_{\text{obj}} = \frac{1}{G} \sum_{i=1}^{G} -[\mathbb{1}(i \in \mathcal{M}) \log(o_i) + (1 - \mathbb{1}(i \in \mathcal{M})) \log(1 - o_i)] \tag{2}$$

We compute the mask loss between the matched masks and the ground truth as

$$L_{\text{mask}} = \frac{1}{N} \sum_{j=1}^{N} \text{BCE}(M_{m_j}^{\text{pred}}, M_j^{\text{gt}}) + \text{DICE}(M_{m_j}^{\text{pred}}, M_j^{\text{gt}}) \tag{3}$$

In practice, we find that computing the BCE only for the points with top $k$ largest loss improves performance, where $k = 512$ for grasping and $k = 1024$ for placing. This is likely due to the large class imbalance (over 90% of the points are not contact points). See Sec. 4.2 for ablations.

The ADD-S loss is introduced by [5] and is critical for good grasp confidence estimation (see Sec. 4.2 for ablations). To compute it, we first need to define 5 key points $\{\mathbf{v}_k\}$ on the gripper. Then, for each pair of predicted grasp and ground truth grasp, we compute the total distance between transformed key points

$$d_{ij} = \sum_{k=1}^{5} \|(R_i^{\text{pred}} \mathbf{v}_k + \mathbf{t}_i^{\text{pred}}) - (R_j^{\text{gt}} \mathbf{v}_k + \mathbf{t}_j^{\text{gt}})\| \tag{4}$$

Next, we find the closest ground truth to each prediction $n_i = \arg\min_j d_{ij}$ and compute ADD-S as

$$L_{\text{ADD-S}} = \frac{1}{|\mathcal{C}^{\text{pred}}|} \sum_{i \in \mathcal{C}^{\text{pred}}} s_i d_{in_i} \tag{5}$$

where $\mathcal{C}^{\text{pred}}$ is the set of contact points of predicted grasps and $s_i$ is the grasp confidence, a scalar between 0 and 1 taken from the contact masks before thresholding. Note that since the loss is weighted by $s_i$, predicted grasps that are far away from any ground truth grasp will get a larger penalty on confidence, which improves contact point estimation.

**Placing:** The placing objective is defined as a combination of BCE and DICE [25] between the predicted and ground truth placement masks

$$L_{\text{placing}} = \frac{1}{P} \sum_{i=1}^{P} \text{BCE}(M_i^{\text{pred}}, M_i^{\text{gt}}) + \text{DICE}(M_i^{\text{pred}}, M_i^{\text{gt}}) \tag{6}$$

This is the only loss for placing since no other learnable quantities are needed to reconstruct the placement poses (see 3.3).

## 3.3 Model Inference

**Grasp Pose Prediction:** During inference, we first select the contact masks whose objectness score is above 0.5. Then, for each point $\mathbf{p}$ within the contact mask, we take the corresponding grasp parameters from the action decoder to reconstruct a 6-DoF grasp pose $(R_{\text{grasp}}, \mathbf{t}_{\text{grasp}}) \in \text{SE}(3)$ as

$$\mathbf{t}_{\text{grasp}} = \mathbf{p} + \frac{w}{2}\mathbf{c} + d\mathbf{a} \tag{7}$$

$$R_{\text{grasp}} = \begin{bmatrix} | & | & | \\ \mathbf{c} & \mathbf{c} \times \mathbf{a} & \mathbf{a} \\ | & | & | \end{bmatrix} \tag{8}$$

where $\mathbf{c}$ is the unit 3D contact direction, $\mathbf{a}$ is the unit 3D approach direction, $w$ is the 1D grasp width and $d$ is the constant distance from the gripper base to the grasp baseline (the line between two fingers). We refer readers to Contact-Grasp-Net paper for more details on the formulation [5].

**Placement Pose Prediction:** The $P$ placement contact masks represent valid placement locations when the object in the gripper is rotated by $P$ discrete planar rotations $R_{\text{planar}}$. To recover the placement poses, we first compute the bottom center $\mathbf{b}$ of the object point cloud which is used as the reference point for contact. Then, we use forward kinematics to obtain the gripper's current pose $(R_{\text{ee}}, \mathbf{t}_{\text{ee}})$. The 6-DoF placement pose $(R_{\text{place}}, \mathbf{t}_{\text{place}})$ can be computed as

$$\mathbf{t}_{\text{grasp}} = \mathbf{p} + R_{\text{planar}}(\mathbf{t}_{\text{ee}} - \mathbf{b}) \tag{9}$$

$$R_{\text{grasp}} = R_{\text{planar}} R_{\text{ee}} \tag{10}$$

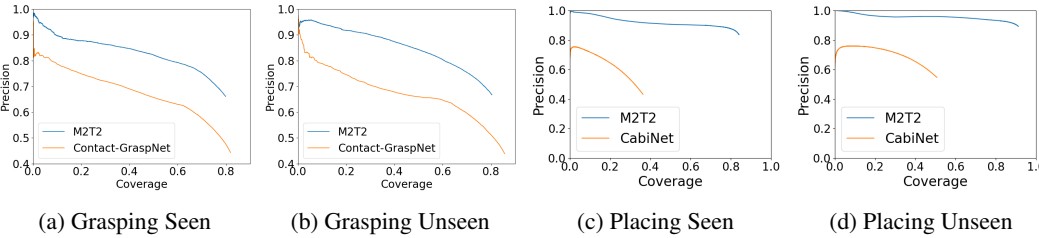

| (a) Grasping Seen | (b) Grasping Unseen | (c) Placing Seen | (d) Placing Unseen |

Figure 3: M2T2 outperforms task-specific models – Contact-GraspNet [5] for grasping and CabiNet [6] for placing – on objects from seen categories (a,c) and unseen categories (b,d).

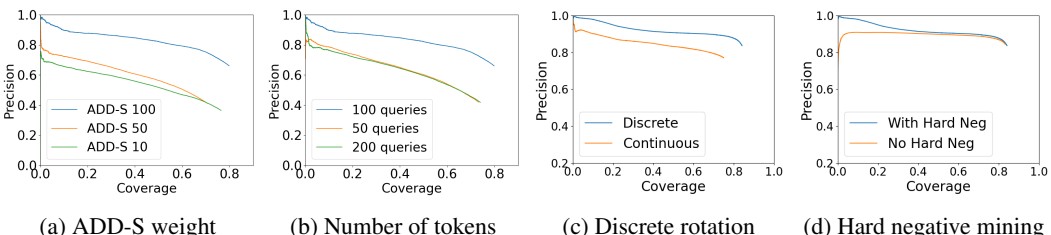

| (a) ADD-S weight | (b) Number of tokens | (c) Discrete rotation | (d) Hard negative mining |

Figure 4: Ablation studies

## 3.4 Synthetic Data Generation

We build a synthetic dataset with 130K cluttered scenes for training and evaluating 6-DoF picking and placing methods. There are 64K training scenes and 1K test scenes for picking and placing each. There are 1 to 15 objects per scene scattered on a randomly sized table mounted with a Franka Emika robot arm. The objects are sampled from the ACRONYM [26] dataset, which contains 8.8K object models, each labeled with 2K grasps. The objects are from 252 different categories, 12 of which are excluded from training. Half of the test scenes contain only objects from the 12 unseen categories. For each scene, we render a $512 \times 512$ depth image from a random viewpoint above the table to generate the scene point cloud. We include example images of the dataset in the appendix.

## 4 Experimental Evaluation

### 4.1 Evaluation in Simulation

**Evaluation Metric:** We use the precision-coverage curve as the metric for our evaluation in simulation. To plot this curve, we start with a confidence threshold of 1 and add grasps/placements to the set of predictions by incrementally lowering the confidence threshold until 0.5. In the mean time, we keep track of two numbers: precision, the percentage of successful grasps/placements, and coverage, the percentage of ground truth grasps/placements that are within 5 cm of any predicted pose. Finally, we plot the coverage on the $x$-axis and precision on the $y$-axis. In practice, we found that this curve is a good indicator of a model's performance in the real world.

A grasp is considered successful if the gripper does not collide with the scene (including occluded parts) and is stable. We evaluate the stableness of a grasp by shaking the grasped object for 5 seconds in the Isaac gym simulator [27] with a *physx* physics engine, identical to the evaluation in ACRONYM [26]. A placement is considered successful if both the gripper and the object are collision-free and the bottom of the object is less than 5 cm from the correct placement surface.

**M2T2 vs. Specialized Baseline Models:** We compare M2T2 against two state-of-the-art specialized models: Contact-GraspNet [5] for grasping and CabiNet [6] for placing. The results are summarized in Fig. 3. M2T2 outperforms both models by a significant margin, especially on the placement. This shows the advantage of orientation reasoning for placement. In many cases, it is not possible to find a good placement pose without rotating the object in hand.

Table 1: Real Robot Experiments Success Rates

| Model | Pick | Place | Place re-orient | Overall |
|---|---|---|---|---|
| M2T2 (ours) | **85.7**% | **72.2**% | **62.5**% | **61.9**% |
| Contact-GraspNet [5] + CabiNet [6] | 76.2% | 56.2% | 25.0% | 42.9% |

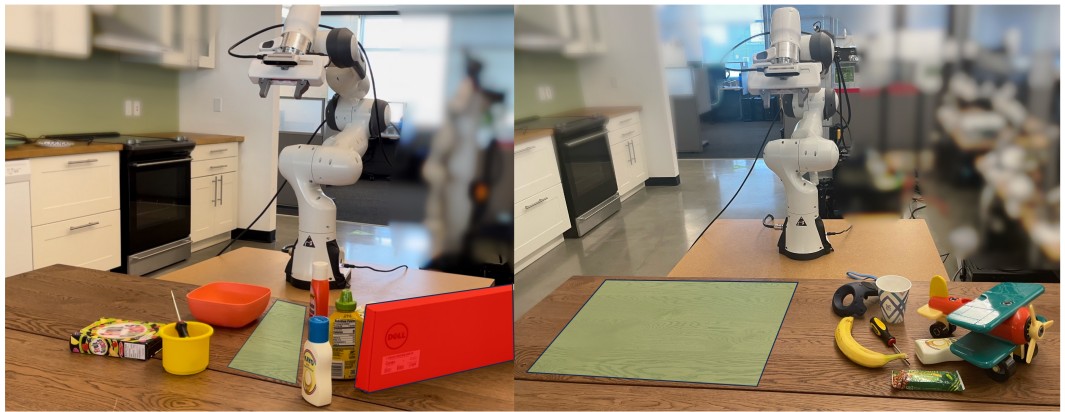

Figure 5: Our robot experimental setup. *Left:* Scenes where the target object (highlighted in red) needs to be reoriented to be placed in the placement region (shown in green). *Right:* A example of a scene where objects are sequentially moved from the right to the initially empty region on the left.

## 4.2   Ablations

**Choice of ADD-S:** We find that setting a larger weight for the ADD-S loss has a critical impact on the grasping performance. As shown in Fig. 4a, setting lower ADD-S increases grasp coverage at the expense of precision which is not desirable.

**Number of Grasp Queries:** We experimented with different number of grasp query tokens and find 100 to be an appropriate number, as shown by the results in Fig. 4b.

**Discrete vs. Continuous Rotation:** We compared our model with a variant where there is only a single placement mask and the placement rotations are regressed just like the grasp parameters. As shown by Fig. 4c, it is better to have a set of placement masks corresponding to discrete rotations. Since multiple orientations of the object can be valid for a given placement location, regressing to a single rotation does not model the multi-modality of placement orientations.

**Importance of Hard Negative Mining:** As mentioned in Sec. 3.2, we use hard negative mining (by applying the mask loss for 1024 points with the largest loss. Without this trick, the quality of most confident placements become significantly worse (see Fig. 4d).

## 4.3   Real Robot Experiments

**Hardware Setup:**   We evaluated M2T2 on a tabletop manipulation setting with a 7-DOF Franka Panda Robot and a parallel jaw gripper. For perception, we used a single Intel Realsense L515 RGB-D camera overlooking the scene. We use the motion planner from [6] for reactive path planning via model predictive control. The picking target and placement region are specified by the user with 3 clicks on the camera image. All inference is run on a single NVIDIA Titan RTX GPU, which takes about 0.1 second per frame. After obtaining a set of collision-free gripper poses from M2T2, the robot execute the one that is closest to the current robot configuration in joint space and has a feasible inverse kinematics solution for the robot arm.

Table 2: Success Rate Comparison on RLBench

| Task | open drawer | turn tap | meat off grill |
|---|---|---|---|
| M2T2 (ours) | **89.3 ± 1.8**% | **88.0 ± 5.6**% | **86.7 ± 1.8**% |
| PerAct [10] | 80.0% | 84.0% | 84.0% |

**Results:** Table 1 shows the success rate of 21 pick-and-place sequences in 7 different scenes. Each scene contains more than 5 objects and all objects are unseen during training. We do not provide 3D models for any of the object. The placing success rate is conditioned on picking success, i.e. overall success = picking success × placing success. We designed four scenes where the object has to be re-oriented before placing to fit in the target region, as shown on the left in Fig 5. We can see that M2T2 significantly outperforms the baseline system, which is a combination of state-of-the-art task-specific models. Notably, M2T2 is 9.5% higher for grasping than [5] and 37.5% higher than [6] for the more challenging re-orientation placement. 2/3 pick failures for M2T2 were for cup objects, which were out-of-distribution during training. 4/5 pick failures for the baseline [5] was because the model did not generate grasps at all or the generated ones were not reachable based on Inverse Kinematics (IK). On the other hand, M2T2 generates grasps with higher coverage and hence have a greater chance of being within the robot's kinematic workspace. We only use 8 bins for the orientation discretization, and further increasing the discretization granularity could potentially reduce our placement error. The real robot executions can be found on the project website.

### 4.4 Evaluataion on RLBench

RLBench [7] is a commonly used benchmark to evaluate multi-task robot manipulation methods. We found that many complex tasks in RLBench can be decomposed into a sequence of primitive actions and solved by M2T2. We demonstrate this by training and evaluating our model on 3 RLBench tasks: open drawer, turn tap and meat off grill. M2T2 is able to outperform PerAct [10], a state-of-the-art multi-task model, on all 3 tasks. The results are summarized in Table 2. We report average success rate over 25 random seeds. The standard deviation is computed with 3 repeated trials for each seed. PerAct's results are taken from the original paper. This demonstrates M2T2's ability to learn action primitives other than generic pick and place and to incorporate multi-modal inputs including language. More details of the experiments are in the appendix.

## 5 Conclusion

In this paper we present Multi-Task Masked Transformer (M2T2), an object-centric transformer model for pick-and-place of unknown objects in clutter. We train M2T2 on a large-scale synthetic dataset of 130K scenes and deploy it on a real robot without any real-world training data. M2T2 outperforms state-of-the-art specialized models in 6-DoF grasping [5] and placing [6] by about 19% in overall success rate in the real world. M2T2 is especially proficient in re-orienting objects for precise collision-free placement. In future, we plan to integrate M2T2 with language-conditioned task planners [3, 4] to build an open-world manipulation system that works on everyday scenes with out-of-distribution objects.

**Limitations:** M2T2's performance is bounded by the visibility of contact points. For example, it cannot predict grasps on the side of the box opposite to the camera. M2T2 is also not able to directly predict actions without contact points, such as lifting. During placing, M2T2 needs segmentation of the object in gripper in order to estimate how far the gripper needs to be from the contact point between the object and placement surface. The grasp predictions for each token can spread across multiple objects in close contact. Currently, M2T2 is trained and evaluated only on tabletop scenes, but this could be improved by training with more diverse procedurally generated synthetic data such as in [6, 28].

**Acknowledgments**

This work is supported by NSF Award IIS-2024057. The project title is Collaborative Research: NRI: FND: Graph Neural Networks for Multi-Object Manipulation.

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

# Appendix

## A Further Architectural Details

**Scene Encoder** The scene encoder is a PointNet++ [23] with 4 multi-resolution set abstraction layers as the encoder and 4 feature propagation layers as the decoder. The input point cloud is sub-sampled to 16384 points. Each set abstraction layer will select $N/4$ seed points using furthest point sampling where $N$ is the size of the input pointwise feature map. Then, local features are computed around each seed point and propagated with an MLP. As a result, the scene encoder produces 4 feature maps of decreasing resolution, with 16384, 4096, 1024 and 256 points respectively. We use the first per-point feature map for prediction and the remaining 3 for cross-attention.

**Contact Decoder** The contact decoder takes $G = 100$ grasp tokens and $P = 64$ placement tokens as input. These input query tokens are randomly initialized and learned during training. The $G + P$ query tokens are fed into a transformer network with 3 blocks. Each block consists of a cross-attention layer, a self-attention layer and a feedforward MLP layer. In the cross-attention layer, the query tokens are cross attended with one of the feature maps produced by the scene encoder to incorporate scene context. In the self-attention layer, the query tokens are attended with themselves to propagate information among different queries. The width (i.e. dimension of each token) is set to 256. The input tokens of each transformer block (including the initial one) are also used to produce intermediate mask predictions. Specifically, the tokens are multiplied with the per-point feature map from the scene encoder, passed through sigmoid and thresholded to generate a per-point mask for each query. These intermediate masks are not only supervised by the ground truth masks during training, but also subsampled and used as attention masks for the cross-attention layer. This forces the network to focus on relevant regions in the scene.

While the contact decoder is inspired by [24], it is specially designed to handle 3D inputs instead of images. For example, since the context features are grounded to 3D points, we compute position encodings from their 3D locations during cross-attention.

**Modifications for RLBench** In RLBench, we break down tasks like "put meat off grill" into predicting a single grasp or placement pose conditioned on the language goal. To make the output conditioned on language, as shown in Fig. A, we can introduce additional language tokens as query tokens, where the language tokens come from the language goal embedded by a frozen CLIP [29] encoder. Following PerAct [10], we trained M2T2 on 100 demos and evaluate on 25 demos with random seeds different from training.

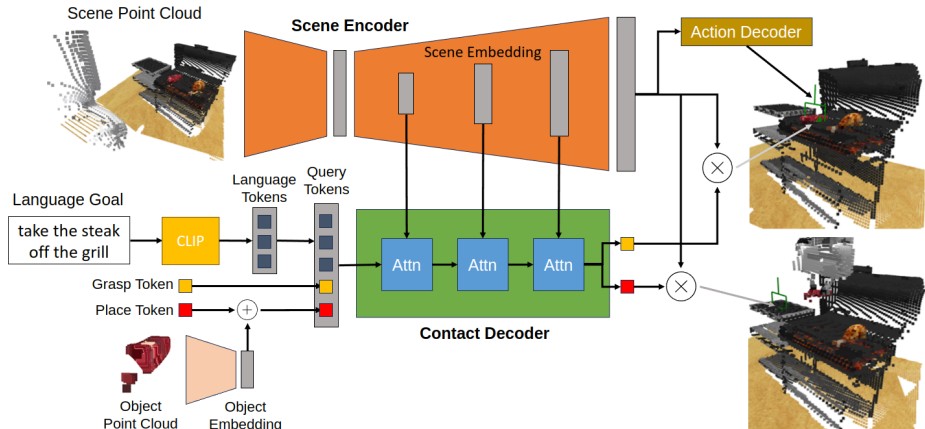

Figure A: Network for language-conditioned pick-and-place in RLBench. Compared to the network for generic pick-and-place, there are only 1 grasp and 1 place query token. The predicted grasp and placement are conditioned on language commands encoded by a frozen CLIP.

# B Comparison Against Single-task Model

We have trained our model to only perform a single task. These task-specialized models are worse than our multi-task model (see Fig. Ba, Bb). This shows the importance to formulate both picking and placing under the same framework.

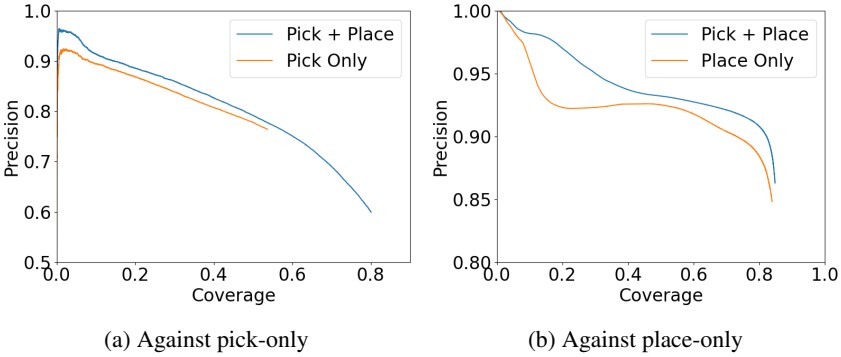

(a) Against pick-only      (b) Against place-only

Figure B: Multi-task vs Single-task Models

# C Training

M2T2 is trained using the Adam-W [30] optimizer with a fixed learning rate of 0.0008 on 8 V100 GPUs for 160 epochs. The batch size is 16 on each GPU. The training takes about 2 days to finish.

# D Data Generation

We procedurally generated a large-scale synthetic dataset for training M2T2, as shown in Fig C. In each scene, we randomly place 1-15 objects from the ACRONYM dataset [26] on the table. Each object in ACRONYM are labeled with 2000 grasps. We transform these grasps by the object pose and filter out colliding ones. The camera pose is randomized around the entire hemisphere above the table, making M2T2 very robust to viewpoint changes.

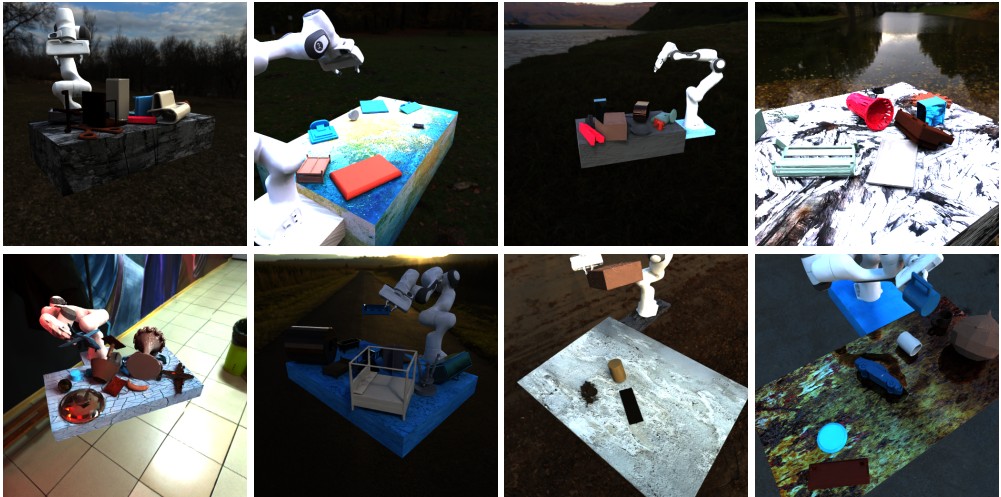

Figure C: Examples for our large-scale synthetic dataset, for the grasping (top) and placing (bottom) tasks respectively. Objects are randomly sampled from ACRONYM [26]. Each scene can contain up to 15 objects, which creates many very cluttered scenes. We also include robot in the observation to simulate realistic occlusion by the robot arm.

