# OpenReview forum: "M2T2: Multi-Task Masked Transformer for Object-centric Pick and Place"
_robot-learning.org/CoRL/2023/Conference — CoRL 2023 Poster_

### Official Review · Reviewer_bMyz · 2023-07-16

**Confidence:** 4
**Originality:** Fair
**Technical Quality:** Fair
**Clarity Of Presentation:** Fair
**Impact:** 3

**Recommendation:**

Weak Reject: I recommend rejecting the paper, but will not argue for my recommendation if the majority of other reviewers have a different opinion.

**Review:**

Strengthes

1. The paper presents a unified transformer model for object-centric, 6-DoF grasping and placing. Experimental results demonstrate that M2T2 outperforms methods Contact-graspnet and CabiNet [7, 8] in terms of success rate and output diversity.

2. The paper collects a large-scale dataset which consists of 164K cluttered scenes and 8.8K different objects.

3. It shows that M2T2 achieves zero-shot sim2real transfer for pick-and-place in unknown scenes and objects, outperforming baseline methods by about 19%.

Weaknesses

1. The paper's characterization of the current gap does not seem to be true. We see general foundation models including Palm-E, RT-1 which are capable to multiple manipulation tasks includes picking, placing, opening and closing drawers, getting items in and out drawers, placing elongated items up-right, knocking them over, pulling napkins and opening jars. These models predict 6-DOF (translation and rotation) and grasping parameters. For example, RT-1 exhibits varying levels of generalization: L1 for generalization to the new counter-top layout and lighting conditions, L2 for additionally generalization to unseen distractor objects, L3 for additional generalization to drastically new task settings, new task objects or objects in unseen locations such as near a sink. It is not clear to the reviewer these models are limited by low-level manipulation primitives.

PaLM-E: An Embodied Multimodal Language Model, https://palm-e.github.io/
RT-1: Robotics Transformer for Real-World Control at Scale, https://robotics-transformer.github.io/

2. The paper claims M2T2 is a unified architecture. However, it seems to be designed separately for picking and placing. Then the two architectures are merged in a straightforward way. This is evident that object encoder is only used for placement tasks and action decoder only applies to picking tasks. Please explain the advantage of M2T2 with respect to general foundation model architectures such as RT1, PerAct, etc.

3.  The baseline methods chosen to compare is not clearly explained. Why combining Contact-GraspNet [7] + CabiNet [8] is the best SOTA? Why not PerAct and other models? It is not clear what contributes to M2T2's performance gain with respect to the baselines.

**Quality Of The Limitations Section:**

Limitations are addressed clearly

**Questions For Rebuttal:**

Please address the three key weaknesses.

Besides, the paper has serious writing, organization, and positioning problems. The intro and related work does not give a clear picture of the proposed method in the context of related work. The architecture is not clearly motivated other than it takes inspiration from mask2former. Is the scene encoder and contact decoder from mask2former the key contribution to performance gains?

=== Post rebuttal
I am slightly more positive given the new results comparing with PerAct. I still think the paper is borderline. Please interpret my score after rebuttal to be in between weak accept and reject. I still think the architecture is very much specialized, and may not be relevant for a system that solves manipulation tasks in practice. Do you really need specific modules for pick only or place only? Why can not pick and place policies be learned without these specialized modules like RT-2? In particular, will these specialized modules hurt out of distribution generalization?

**Robotics Focus:**

Sufficient demonstration on hardware

**Summary Of Paper:**

The paper argues that there are two recent bodies of work. The first is to leverage Large Language Models (LLMs) and large-scale robotic datasets for high-level decision-making for object manipulation. However, these generic models are limited in terms of low-level pick-and-place primitives. The second is specialized models excel in low-level manipulation of unknown objects but only work for a single task. The paper claims that there is a gap to bridge. It proposes Multi-Task Masked Transformer (M2T2) which intends to achieve the goal of pick and place arbitrary objects in cluttered scenes. M2T2 adopts a transformer model which predicts pick and place poses for all (unknown) objects given just a point cloud observation of the scene. M2T2 is trained with a large-scale synthetic dataset consisting over 160K scenes and evaluated on zero-shot sim2real transfer on the real robot.

**Summary Of Recommendation:**

The paper's motivation is questionable, does not seem to be true. The architecture is also not clearly motivated. The baseline methods chosen are not clearly explained.

---

### Official Review · Reviewer_whqu · 2023-07-19

**Confidence:** 4
**Originality:** Very Good
**Technical Quality:** Good
**Clarity Of Presentation:** Good
**Impact:** 3

**Recommendation:**

Weak Accept: I recommend accepting the paper, but will not argue for my recommendation if the majority of other reviewers have a different opinion.

**Review:**

Strengths:
- A single model that predicts both pick and place skills.
- Provides a scalable pipeline in simulation to train robot policies.

Weakness:
- The first paragraph in Related Works is confusing. How is it relevant to this paper? Why do the authors need to have a specific paragraph when there is also a paragraph on multi-task learning for robotics? No connection is made between this paragraph and the proposed work
- The task definitions are not self-explanatory. The videos also only include limited motion or behaviors. Please provide more information/details in the store.
- This paper requires more explanation on how the learned models would be used in task executions.
- The experiments need more ablations on how each design choice impacts the policy performance. The current status totally misses such a part.
- Additional comments: It was not very clear whether it uses RGB-D or Depth point clouds. Figure 1 is quite misleading, given that the models did not use any RGB information. I think it’s better to either display depth images in Figure 1 or add captions to clarify that only depth is used in the following model design.
- The authors chose to use a discrete set of rotations and claim that continuous rotations will have the issue of regressing to multi-modality (Ln204 - Ln 208). But surely, a model can regress to multiple values.

**Quality Of The Limitations Section:**

Limitations are addressed clearly

**Questions For Rebuttal:**

- Please address the points mentioned in the weakness above
- How is the ability of the model to perform a very long horizon task of placing? What kind of complicated tasks can this model handle?
- Is this model able to perform tasks where objects might be placed on planes of different heights (for example, a shelf)?
- What does “Masked Transformer” mean in the title? It is confusing which “masks” this title refer to.


**Robotics Focus:**

Sufficient demonstration on hardware

**Summary Of Paper:**

The authors proposed a transformer-based approach to learning multi-task pick and place. In order to learn such a model, the authors introduce a synthetic data generation pipeline to train the policies.


**Summary Of Recommendation:**

Overall the proposed approach makes sense, and experiments are provided. However, the video demonstration of placing one object specifically in a position seems to sell the model short. Since the proposed model is targeted at performing pick-and-place seamlessly, I think the authors need to demonstrate how the learned model is used to perform a long horizon task of setting the table. Many questions need to be clarified before this score remains or improves.

---

### Official Review · Reviewer_dMwG · 2023-07-29

**Confidence:** 4
**Originality:** Very Good
**Technical Quality:** Very Good
**Clarity Of Presentation:** Very Good
**Impact:** 3

**Recommendation:**

Weak Accept: I recommend accepting the paper, but will not argue for my recommendation if the majority of other reviewers have a different opinion.

**Review:**

Overall the paper has quite a lot of noteworthy strengths. First, the method is fairly general: a unified transformer model for object-centric 6 DoF grasping and 4 DoF placing that is based on raw 3D RGB-D point clouds. The architecture and technical approach (as described in section 3) is intuitive, elegant, and well-explained. Essentially the paper leverages existing motion generation approaches and transforms the pick-and-place task from a “robotics” task to a “3D scene understanding” task. When trained with large-scale dataset and supervised learning (with well-designed losses, as described nicely in section 3.3), the model seems to perform pretty well.

Furthermore, the experimental evaluation both in sim and in real are fairly solid. The proposed method is compared against the respective state-of-the-art models for picking and placing and achieves better results. It’s also noteworthy that the proposed method achieves sim2real transfer in a zero-shot manner. In fact, the real experiments are very challenging because each scene contains more than 5 objects and all objects are unseen during training. Although not perfect, the proposed method also beat baselines by a significant margin in the real experiments. Finally, the ablation studies in 4.2 are helpful for readers to understand which part of the algorithmic design matters.

In terms of rooms for improvement, the paper can benefit from including more failure case analysis in both sim and real. Also, it would be helpful if the authors can quantify the sim2real gap when performing zero-shot transfer. Right now the sim results and the real results are completely separate and it’s difficult for researchers to draw lessons from them. Figure 2 can probably be improved to be more self-contained. In other words, as for now, Figure 2 is very difficult to understand without reaching section 3.1 in detail. Figure 4 can also be improved (the images are dark and a bit distorted). Maybe also consider adding another figure that is analogous to Figure 4 for placing. Also, the supplementary video is faulty (a large portion of the video is static, e.g. stuck starting from 0:44 for instance). Finally, I have compiled a list of clarification questions for the paper authors below. It would be great if the authors could help clarify (at least some of) those points in the revised manuscript.



**Quality Of The Limitations Section:**

Limitations are addressed clearly

**Questions For Rebuttal:**

- Why are the two task embeddings needed, if you already have grasp embeddings [GxD] and place embeddings [PxD]?
- How is the grasp confidence s_{i} predicted? Is it the same as M_{i}?
- During real-world model inference, for a given object, is the grasp with the highest confidence being executed? If it fails, does the robot retry?
- During training, do the groundtruth grasps/placements consist of zeros and ones, or are the ground truth grasps/placements also annotated with confidence?
- On a related note, how are the ground truth grasps for each object generated? Is clutter taken into account? For example, if a cup is placed closely next to another object on the right hand side, are the grasps from the right hand side annotated as invalid (because the gripper would collide with that other object)? Same question applies to the ground truth placements.
- If the answer to the above question is Yes, which means the dataset takes clutter into account, what about the whole trajectory of the robot arm to reach the desired gripper pose? In other words, for each positive label of ground truth grasp in the dataset, is it guaranteed that there is an entire trajectory for the robot to accomplish that grasp without colliding with any object in the scenes. For example, in the supplementary video, the robot tries to place the bowl next to the cracker box. Even though the bowl and the gripper don’t collide with the cracker box in the end configuration, the robot arm actually collides with the cracker box during the execution. How would you handle cases like these?
- In the limitation section, it’s mentioned that “Additionally, the grasp predictions for each token can spread across multiple nearby objects even though during training, we want each pick token to predict grasps corresponding to only one object.” Can you address this with a better design of loss function?


**Robotics Focus:**

Sufficient demonstration on hardware

**Summary Of Paper:**

The paper presents Multi-Task Masked Transformer (M2T2), a solution for picking and placing arbitrary objects in cluttered scenes (which might require careful reorientation of the grasped objects before placing them). The method is pretty general in the sense that it takes raw RGB-D point clouds as input with no known object models. It’s trained with a large scale synthetic dataset and achieves zero-shot sim2real transfer  on the real robot. It significantly outperforms the state-of-the-art pick and place methods, respectively, especially in challenging scenes where the objects need to be reoriented for collision-free placement with precision.


**Summary Of Recommendation:**

Based on the review above, I recommend Weak Accept. I am open to changing my mind during the rebuttal.

---

### Official Review · Reviewer_Bsjg · 2023-07-31

**Confidence:** 3
**Originality:** Good
**Technical Quality:** Good
**Clarity Of Presentation:** Good
**Impact:** 3

**Recommendation:**

Weak Accept: I recommend accepting the paper, but will not argue for my recommendation if the majority of other reviewers have a different opinion.

**Review:**

The paper tackles an important problem in robotic object manipulation: Finding grasp and placement poses. It proposed a novel technique that learns to predict these poses using a single model, and by that clearly differs from previous work. However, it lacks clarity and the experiment section could be improved.

Strength:
+ learns joint representation for picking and placing
+ generates a set of possible grasps/placements in one forward pass
+ proposed method is easily scalable to large training datasets
+ consider placement position and planar rotation
+ experiments show generalization to unseen objects

Weakness:
- paper lacks clarity at some points (see questions)
- unclear whether the comparison to the state-of-the-art methods is meaningful (see questions)
- imo further comparisons to baselines would be nice (see questions)
- imo further ablation studies would be interesting (see questions)

**Quality Of The Limitations Section:**

Additional details required

**Questions For Rebuttal:**

Technical Approach:
- Why is the query embedding formed using a task embedding and grasp/place embedding? What is the Motivation for this, e.g. why does the task embedding exist?
- How are the different losses combined/weighted?
- How is the object point cloud computed? If some segmentation model is used, imo this should be addressed in the limitations section.
- More details on architecture hyperparameters, e.g. #neurons in the MLPs, would be nice for reproducability (supplementary material)

Experiments:
- You ablate the importance of hard negative mining, however, in the technical approach section hard negative mining is never mentioned.
- It is unclear to me whether the comparisons to the baselines is fair: Are the baselines retrained on the same dataset as your model? How do they compare in model capacity to your model? Is your model the first that predicts placement poses with planar rotation from point clouds and, if not, why not compare to a method that also considers planar rotation?
- Would be interesting to investigate/ablate the effects of the multi-task setting, e.g. train the architecture only for grasping or placing. How much of the performance gain comes from the architecture (which differs from the compared baselines)? How much from the multi-task training?
- Would be interesting to report inference time. How does inference time compare to the inference time of the baselines?
- The model predicts multiple grasp and placement poses. For the real robot experiments, how is decided which of these grasp and placements are executed for one pick-and-place sequence?

**Robotics Focus:**

Sufficient demonstration on hardware

**Summary Of Paper:**

The authors propose a multi-task transformer model that predicts contact masks for picking and placing from a 3D point cloud of the scene.
These contact masks are then used to compute grasp and placement poses.
The model is trained end-to-end using a synthetic dataset with ground-truth information.
The authors evaluate the model and several design choices in simulation as well as its zero-shot sim2real performance.
Their experimental results indicate that the proposed method outperforms state-of-the-art single-task baselines.

**Summary Of Recommendation:**

The proposed method is novel, relevant and technical sound. However the paper lacks clarity and there are questions that should be addressed. Therefore I am recommending a weak reject.

---

### Author Response · Authors · 2023-08-16
**Response to All Reviewers**

We thank all the reviewers for carefully reading our paper and providing valuable feedback. We are encouraged by comments such as "this paper tackles an important problem in robotics" and "the experimental evaluation both in sim and in real are solid". We acknowledge that there is room for improvement and we are making continuous effort to improve the paper. We would like to highlight two noteworthy experiments that adds more significance to our work.

**Comparison against PerAct on RLBench**

RLBench [1] is a commonly used benchmark to evaluate multi-task robot manipulation methods. We compared M2T2 against PerAct [2], a state-of-the-art multi-task model, on 3 RLBench tasks and shows that M2T2 achieves better performance on all 3 tasks. We report average success rate over 25 seeds per task and standard deviation over 3 trials per seed.

Task   | open drawer | turn tap   | meat off grill
------ | ----------- | ---------- | --------------
PerAct | 80          | 84         | 84
M2T2   | **89.3 ± 1.8**  | **88.0 ± 5.6** | **86.7 ± 1.8**

Although there are many tasks in RLBench, most of them can be decomposed into pick-and-place sequences. We picked 3 tasks due to time constraint, but M2T2 can be applied to any task where the contact points for picking and placing actions are visible.

**Multi-task vs single-task models**

We have trained our model to only perform a single task: grasping or placing. These task-specialized models are worse than our multi-task model. This shows that it is important to formulate both picking and placing under the same framework. The precision-coverage curve comparisons can be found in the attached files in the rebuttal.

**Attachments and Website**

In the attached zip file to each rebuttal, there are 4 figures:
- Updated architecture figure without task embedding
- Architecture figure for model trained on RLBench
- Precision-coverage curve against picking-only model
- Precision-coverage curve against placing-only model

You can also find videos for all the real-world evaluations on our anonymous project website https://m2-t2.github.io. Please check them out!

[1] James, Stephen, et al. "Rlbench: The robot learning benchmark & learning environment." IEEE Robotics and Automation Letters 5.2 (2020): 3019-3026.

[2] Shridhar, Mohit, Lucas Manuelli, and Dieter Fox. "Perceiver-actor: A multi-task transformer for robotic manipulation." Conference on Robot Learning. PMLR, 2023.

---

### Decision · Program_Chairs · 2023-08-30

**Decision:**

Accept (Poster)

**Comment:**

The paper proposes an unified transformer based model for predicting grasp poses and placements for unknown objects in cluttered scenes.
While overall the presented method and ideas – unified transformer based model , the large-scale training (and dataset) is appreciated, and real word experiments are provided, there are mixed feeling among the reviewers. The main critic points are related to the comparison (and definition) of the state-of-the-art comparisons, ablation studies, additional experiments, and missing details on the tasks, execution, model, etc. During the rebuttal, the authors tackled most of the raised concerns and provided additional requested evaluations. The general consensus tends to accept the paper, but the authors need to update the manuscript with the additional evaluations and clarifications as well (taking the page limit into account!).